# Thymoquinone Pectin Beads Produced via Electrospray: Enhancing Oral Targeted Delivery for Colorectal Cancer Therapy

**DOI:** 10.3390/pharmaceutics16111460

**Published:** 2024-11-15

**Authors:** Mulham Alfatama, Hazem Choukaife, Okba Al Rahal, Nur Zahirah Mohamad Zin

**Affiliations:** 1Faculty of Pharmacy, Universiti Sultan Zainal Abidin, Besut Campus, Besut 22200, Malaysia; hazemchoukaife@gmail.com; 2School of Chemistry, University of Birmingham, Edgbaston, Birmingham B15 2TT, UK; okba.alrahal@gmail.com

**Keywords:** thymoquinone, pectin beads, colorectal cancer, colon targeting, electrospray

## Abstract

**Background/Objectives**: Thymoquinone (TQ) exhibits diverse biological activities, but its poor solubility and bioavailability limit its cancer efficacy, requiring innovative solutions. This study explores the development of an oral delivery system targeting colon cancer based on TQ pectin beads (TQ-PBs) produced through an adjustable electrospray technique. This study hypothesised that adjusting bead diameter through the electrospray technique enables precise control over water absorption and erosion rates, thereby achieving a controlled release profile for encapsulated TQ, which enhances targeted delivery to the colon. **Methods**: TQ-PBs were synthesised and optimised using an electrospray technique based on the ionic gelation method. The prepared beads were characterised based on particle size, sphericity, encapsulation efficiency (EE), water uptake, erosion, surface morphology, molecular interactions, and texture. The cumulative TQ release studies, an accelerated stability test, and cytotoxicity evaluation against the colon cancer HT-29 cell line were also assessed. **Results**: The optimised TQ-PB formulation demonstrated an average bead size of 2.05 ± 0.14 mm, a sphericity of 0.96 ± 0.05, and an EE of 90.32 ± 1.04%. The water uptake was 287.55 ± 10.14% in simulated gastric fluid (SGF), 462.15 ± 12.73% in simulated intestinal fluid (SIF), and 772.41 ± 13.03% in simulated colonic fluid (SCF), with an erosion rate of 45.23 ± 5.22%. TQ release was minimal in SGF (8.13 ± 1.94% after 2 h), controlled in SIF (29.35 ± 3.65% after 4 h), and accelerated in SCF (94.43 ± 2.4% after 3 h). Stability studies over one month showed a size reduction of 17.50% and a 6.59% decrease in TQ content. Cytotoxicity assessments revealed significant anticancer activity of TQ-PB, with an IC50 of 80.59 ± 2.2 μg/mL. **Conclusions**: These findings underscore the potential of TQ-PB as an effective oral drug delivery system for targeted colorectal cancer therapy.

## 1. Introduction

Colorectal cancer (CRC) is recognised as the third most prevalent cancer worldwide and is characterised by its multifactorial aetiology, which encompasses genetic, environmental, and lifestyle factors [1]. This gastrointestinal malignancy typically arises from polyps—abnormal growths within the inner lining of the colon. The pathogenesis of CRC is complex, involving a confluence of factors, including genetic predisposition, dietary habits, and inflammatory conditions affecting the gastrointestinal tract. The progression of CRC is driven by intricate molecular processes, including unregulated cellular proliferation, enhanced angiogenesis, and the dysregulation of programmed cell death mechanisms, such as autophagy, apoptosis, and necroptosis [2]. Despite the lack of a definitive cure for CRC, various treatment modalities are available for disease management, including surgical resection, radiation therapy, and chemotherapy. Surgical intervention remains the most frequently employed and preferred approach. Chemotherapy utilises a range of agents, including doxorubicin, fluorouracil, cisplatin, and leucovorin, aimed at targeting neoplastic cells. However, these conventional treatments are frequently associated with significant adverse effects, such as nausea and gastrointestinal complications, attributable to their inherent toxicity and limited specificity for cancer cells [3]. Novel therapeutic strategies and more effective treatments for advanced CRC remain critical priorities.

Nigella sativa oil, commonly referred to as black seed oil (BSO), has garnered substantial interest in recent studies due to its diverse therapeutic properties. It is extracted from the seeds of Nigella sativa, or black cumin, possessing antimicrobial [4], anti-inflammatory [5], anticancer [6], antidiabetic activities [7], and antihypertensive [8] properties. Thymoquinone (TQ) is a naturally occurring compound renowned for its diverse protective properties, including antioxidative, anticancer, immunomodulatory, and antimicrobial activities. As the principal bioactive component of black seed volatile oil, TQ demonstrates significant potential in inducing apoptosis and modulating the expression of both pro-apoptotic and anti-apoptotic genes. Furthermore, it plays a crucial role in inhibiting cancer metastasis by activating the c-Jun N-terminal kinase (JNK) and p38 signalling pathways [9]. TQ has demonstrated its anticancer activity against CRC in numerous studies [10]. Despite its potent anticancer effects, TQ is hindered by low bioavailability resulting from poor aqueous solubility and stability, complicating its administration and targeting of cancer cells. Thus, innovative strategies are crucial for enhancing the bioavailability of plant-derived compounds with significant anticancer potential. Various strategies have been employed in cancer treatment, particularly through drug delivery systems featuring particle sizes ranging from nanometres to millimetres [11,12,13,14]. A study encapsulated TQ in lipid-polymeric nanoparticles composed of phosphatidylcholine (PC) and poly(lactide-co-glycolide) (PLGA) to enhance its anticancer potency against the Caco-2 cell line and oral delivery effectiveness [15]. Another study delivered high concentrations of TQ directly to the colon for treating inflammatory bowel disease via oral administration using alginate microcapsules [16].

The oral route of administration is a widely accepted and straightforward method for drug delivery, providing numerous advantages, including non-invasiveness, enhanced patient compliance, the ability for self-administration, and cost-effectiveness. An optimal orally delivered medication for CRC treatment should effectively reach the colon and rectum, specifically target lesion sites for accumulation, and also possess the capability to address metastatic lesions through systemic circulation [17]. One of the primary factors limiting the use of oral administration is the concern related to stomach acidity, enzymatic degradation, and poor bioavailability. Therefore, the development of an effective oral delivery system for colon targeting necessitates several strategies, including minimising premature drug release in the stomach and small intestine [18].

Pectin, a natural carbohydrate polymer found in the cell walls of most plants, holds significant promise for targeting CRC through drug delivery systems. This potential arises from its non-toxicity, gelling capabilities, mucoadhesive properties, efficient digestibility by colonic microbes, and anticancer activity [19]. The physicochemical characteristics of pectin facilitate ionotropic gelation, allowing it to interact with oppositely charged inorganic ions such as calcium and zinc cations to form insoluble three-dimensional hydrogel networks in an acidic medium, which are also digestible by colonic microbes. Furthermore, pectin can interact with oppositely charged ionic polymers to produce polyelectrolyte complexes (PECs), thereby enhancing its functional properties [20]. The mucoadhesive attribute of pectin can enhance the residence time of the anticancer agents in the colon region, thereby improving their therapeutic efficacy and bioavailability [21]. Additionally, the controlled release of both hydrophobic and hydrophilic drugs through the pectin delivery system across the gastrointestinal tract is crucial in targeting the colon while minimising drug loss [22]. An insulin-loaded trimethyl chitosan nanoparticle coated with pectin was recently developed for oral delivery targeting the colon, aiming to enhance the effectiveness of oral insulin through its improved colonic absorption [23].

In this study, pectin polymer was utilised to form hydrogel beads as a colonic drug delivery system by oral administration. A variety of techniques are available for encapsulating bioactive compounds within hydrogel beads [24]. Among these methods, electrospraying is particularly notable for its advantages as a single-step process that does not require organic solvents. This technique utilises a focused electric field to regulate the size of drug carriers by overcoming surface tension through electrostatic forces [25]. 

Based on previous research, we hypothesised that utilising the electrospray technique to fabricate pectin beads with precisely controlled sizes could enhance their behaviours in the gastrointestinal tract. By adjusting the bead diameter, we aimed to modulate water absorption and erosion rates, thereby achieving a controlled release profile for encapsulated TQ that enhances targeting specifically to the colon. BSO, a natural source of thymoquinone with a TQ solubility exceeding 500 mg/mL, was employed as a solvent to replace toxic organic solvents and increase the drug loading capacity of the developed TQ-PB.

This study aimed to develop an oral dosage form of oral TQ colonic delivery for colorectal cancer (CRC) treatment. TQ was loaded within a cross-linked pectin matrix using the electrospray technique based on ionic gelation, producing TQ-pectin beads (TQ-PBs). The formulation was optimised using central composite design (CCD) to refine key parameters, including particle size; sphericity; water uptake in simulated gastric fluid (SGF), simulated intestinal fluid (SIF), and simulated colonic fluid (SCF); and erosion. The pectin beads were designed to protect TQ from degradation in the harsh pH environment of the upper gastrointestinal tract, thereby ensuring its stability until it reached the colon. Once in the colon, the pectin beads disintegrated, releasing TQ at the site of action, where it interfered with the carcinogenic processes of cancer cells.

## 2. Materials and Methods

Thymoquinone and pectin citrus with a molecular weight (MW) > 100 kg/mol (Product No. 26234-05) were purchased from Sigma Aldrich, Taufkirchen, Germany, and Nacalai Tesque, Kyoto, Japan, respectively. Calcium chloride granules (Fisher Scientific, Waltham, MA, USA) were used as a cross-linking agent. RPMI 1640 (with glucose, glutamine, and phenol red), penicillin-streptomycin mixed (Nacalai Tesque, Waltham, MA, Kyoto, Japan), foetal bovine serum (FBS), and phosphate-buffered saline (Sigma Aldrich, Taufkirchen, Germany) were used as cell culture reagents. MTT kit (Sigma Aldrich, Taufkirchen, Germany) was used for the cytotoxicity assay.

### 2.1. Selecting the Independent and Dependent Variables for Optimisation of Pectin TQ-Beads (TQ-PB)

A Design of Experiment (DoE) approach was utilised to assess the statistical significance of both individual and combined effects of formulation variables on experimental outcomes. CCD was implemented to minimise the number of experiments required for optimisation. Design Expert software version 13 (Stat-Ease, Inc., Minneapolis, MN, USA) was used for statistical analysis and optimisation. In the CCD, a four-factor design with six center point trials was incorporated to assess the reproducibility of the method applied. Four independent variables were investigated as follows: the pectin concentration (A), voltage (B), BSO ratio (C), and CaCl_2_ concentration (D), as outlined in Table 1. The dependent variables (responses) measured included particle size (R1, mm), sphericity (R2, *w*/*w*), encapsulation efficiency (EE) (R3, %), water uptake SGF (R4, %), water uptake SIF (R5, %), water uptake SCF (R6, %), and erosion (R7, %). A total of 54 experimental runs were proposed by the CCD to assess these responses. The optimal model for each response was selected based on key statistical parameters, such as the regression coefficient (R^2^), *p*-value, and lack of fit. Furthermore, one-way analysis of variance (ANOVA) with a 95% confidence interval was performed to determine statistical differences between multiple groups and to assess the significance of each independent variable. The optimal formulation was meticulously selected based on critical factors such as minimising sphericity and water uptake in SGF and SIF while maximising EE and water uptake in SCF.

### 2.2. Preparation of TQ-PB

TQ-loaded pectin beads were prepared using the electrospray technique based on the ionic gelation method, as described previously [26] and illustrated in Figure 1. The setup consisted of a high-voltage power supply, a syringe pump, a syringe equipped with a stainless-steel needle, and a cross-linking bath (Figure 1) [27]. Initially, an aqueous pectin solution containing 2.5% (*w*/*v*) of Tween 80 was prepared in distilled water, with continuous stirring maintained overnight. Next, TQ dissolved in 1.5 mL of black seed oil (BSO) was added dropwise to 10 mL of the pectin solution while stirring magnetically until a homogeneous mixture was obtained. This mixture was subsequently delivered through a syringe pump (Model: 1000-US/SyringeONE, New Era Pump Systems, Farmingdale, NY, USA) at a constant flow rate of 0.5 mL/min. The solution was electrosprayed into the cross-linking bath by applying a high voltage to the stainless-steel needle (24 G), positioned 5 cm from the collector. The beads underwent gelation under gentle stirring at 250 rpm for 30 min at room temperature. The cross-linked beads were subsequently collected, washed twice with distilled water, and air-dried at 40 °C.

### 2.3. Size and Sphericity

The size of the beads was determined by capturing digital images with a camera and analysing them using ImageJ software version 1.46 r (National Institutes of Health, Bthesda, MD, USA). The length and width of 20 beads were measured, and the average of these two dimensions was used to calculate the size. Sphericity, an indicator of the roundness of the beads, was estimated by measuring the maximum and minimum diameters of 20 beads [28]. The sphericity was calculated using Equation (1).
(1)$$Sphericity=\frac{d_{max}}{d_{min}}$$
where d_max_ and d_min_ represent the maximum and minimum diameters of the beads, respectively. A sphericity value of 1 indicates a perfectly symmetrical and spherical microbead.

### 2.4. Encapsulation Efficiency

The encapsulation efficiency (EE%) of TQ pectin beads (TQ-PBs) was determined indirectly by measuring the amount of TQ present in the cross-linking bath. Following the cross-linking process, the CaCl_2_ solution was collected, filtered using a 0.22 µm syringe filter (Nylon, Shimadzu, Kyoto, Japan), and mixed with methanol in a 1:1 *v*/*v* ratio prior to analysis. The analysis was performed using a Shimadzu LC-20AT HPLC system (Shimadzu, Japan), measuring absorbance at λ_max_ = 254 nm, as per a previously established method [29,30]. The mobile phase comprised acetonitrile and water (70:30, *v*/*v*), pumped at a flow rate of 1 mL/min through an Inspire C18 analytical column (4.6 × 250 mm, 5 µm). Each sample was injected at a volume of 20 µL, with a run time of 20 min. The concentration of TQ (mg/mL) was calculated based on a standard calibration curve developed using a series of TQ concentrations (0.01, 0.05, 0.1, 0.15, and 0.3 mg/mL). The resulting standard curve equation, y= 0.398x + 0.0107, with a coefficient of determination (R^2^ = 0.987), was applied to calculate the TQ concentration. The EE% was calculated using Equation (2).
(2)$$EE=\frac{Initial\;TQ\;amount-untrapped\;TQ\;amount\;}{Initial\;TQ\;amout}\times 100\%$$

### 2.5. Water Uptake and Erosion

The water uptake behaviour of the TQ-PB was studied under three different conditions as follows: SGF (pH of 1.2 ± 0.1) for 2 h, SIF (pH of 6.8 ± 0.1) for 4 h, and SCF (pH of 7.4 ± 0.1) for 3 h (fluids were prepared according to standard protocols). SGF was prepared by dissolving 0.32 g of pepsin enzyme, 0.2 g of sodium chloride, and 0.7 mL of concentrated HCl in 100 mL of distilled water, with the pH adjusted to 1.2 ± 0.1 using 1 M HCl [31]. SIF was prepared by dissolving 0.68 g of KH_2_PO_4_ and 1 g of pancreatin enzyme, adding 8 mL of 0.2 M NaOH to 100 mL of distilled water, and adjusting the pH to 7.4 ± 0.1 with 0.1 M NaOH [31]. SCF was prepared by dissolving 60 mg of pectinase enzyme in phosphate buffer, adjusted to a pH of 7.4 ± 0.1 [32]. For the experiment, 10 g of dried TQ-PB was immersed in 20 mL of each mimic fluid, weighed at different time intervals, and transferred sequentially to the next fluid. After completing the water uptake study, the erosion percentage of the beads was calculated by drying the beads overnight in a lab oven at 40 °C and recording the final weight. The indices of water uptake (WUI) and erosion (EI) were calculated using Equations (3) and (4).
(3)$$WUI=Wt-\frac{Wt\left(d\right)}{Wt\left(d\right)}\times 100\%$$
where Wt = wet weight of bead at t and Wt(d) = dry weight of bead collected at t.
(4)$$EI=Wi-\frac{Wt\left(d\right)}{Wi}\times 100\%$$
where Wi = initial dry bead weight.

### 2.6. Morphology

The surface morphology of the TQ-PB was assessed using a scanning electron microscope (SEM; Quanta FEG 450, FEI, Eindhoven, The Netherlands) at an accelerating voltage of 3 kV with magnifications of 250× and 1000×. Before imaging, the beads were mounted onto metal stubs and coated with a thin layer of gold in a vacuum using a gold sputter coater (Q150R S model, Quorum, Hertfordshire, UK).

### 2.7. Fourier Transform Infrared Spectroscopy (FTIR)

FTIR spectra were conducted for various samples, including the raw materials and the optimised TQ-PB formulation, across the spectral range of 4000–400 cm^−1^. A background spectrum was first recorded and subtracted from the test spectra to ensure precision. The analysis was conducted using a FT-IR spectrophotometer IRPrestige-21 Shimadzu (Tokyo, Japan), with spectral data processed using LabSolutions IR software (version 5.53). The resulting absorption spectra were compiled, and key functional groups were identified. These were compared against reference libraries and literature to confirm the identities of components and assess potential interactions.

### 2.8. Powder X-Ray Diffraction (PXRD) Measurements

To confirm the amorphous state of the prepared beads, PXRD measurements were performed on TQ, blank PB, and TQ-PB using a Rigaku Miniflex II diffractometer. The samples were continuously irradiated with Cu Kα radiation at a scan speed of 2°/min, a step size of 0.02°, and a 2θ range of 6–80°.

### 2.9. Texture Analysis (Recovery Rate and Elasticity Limit)

The texture profiles of the fresh beads were analysed using a TA.XT Plus texture analyser (Stable Micro Systems, Godalming, UK), following previous reported methods [33]. A flat-ended probe was used to compress the beads. Beads were compressed to final deformations ranging from 10% to 50% of their original heights using a load cell and a cylindrical probe (P/35R, Stable Micro Systems, Surrey, UK) at a speed of 10 mm/s at an ambient temperature of 25 ± 1 °C, with a 15 min interval between each deformation test. The probe was programmed to return to its original position immediately after each compression. Force (F) versus time (s) data collected from these tests were used for subsequent analysis. To ensure statistically representative results, 30 samples of TQ-PB beads in both dry and wet (immersed in water in a polypropylene container) forms were compressed in each test.

### 2.10. In Vitro Drug Release and Kinetic Release Study of TQ-PB

The release profile of TQ-PB was assessed in three simulated fluids, namely SGF (pH 1.2 ± 0.1), SIF (pH 6.8 ± 0.1), and SCF (pH 7.4 ± 0.1), until the complete release of the entrapped TQ. These fluids were prepared according to the protocol described in the water uptake study. A measured amount (10 g) of dry beads was placed in a dialysis bag (MWCO 12 kDa; Sigma-Aldrich) and immersed in 20 mL of SGF for 2 h, followed by transfer to SIF for 4 h, and finally into SCF until the complete release of TQ was achieved [34,35].

The release study was conducted in a shaking water bath maintained at 37 °C with controlled agitation at 50 strokes/min. At specific time intervals, 1 mL of the dissolution medium was withdrawn and replaced with fresh medium to monitor the percentage of TQ released. The amount of released TQ was quantified by filtering the collected samples through a 0.22 µm nylon syringe filter (Shimadzu, Japan) and subsequently mixing them with HPLC-grade methanol in a 1:1 volume ratio. The analysis was conducted using a Shimadzu LC-20AT HPLC system (Shimadzu, Japan), with absorbance measured at λ_max_ = 254 nm, following the previously established HPLC method. The cumulative release of TQ (%) was calculated using Equation (5).
(5)$$TQ\;release=\frac{Wt-W}{Wt}\times 100\%$$
where W is the amount of TQ released and Wt is the theoretical total TQ in 10 g microspheres.

The release kinetic of TQ-PB was studied by following zero-order, first-order, the Higuchi equation, Korsmeyer–Peppas, and Hixson–Crowell models.

### 2.11. Accelerated Stability Test

A short-term stability study of TQ-PB was performed in accordance with the International Council for Harmonisation (ICH) guidelines on stability testing. Freshly prepared beads were stored in semi-permeable polyethylene containers under controlled conditions of 40 °C ± 2 °C and 25% relative humidity (RH) ± 5% RH. At the end of one month, the beads were evaluated for particle size, morphology, and EE% to detect any significant changes in their physical properties or drug content over time [36].

### 2.12. Determination of Cytotoxicity of TQ-PB

The cytotoxicity of the prepared TQ-PB was assessed against human colorectal adenocarcinoma cell lines (HT-29) using the MTT assay over a 24 h period. HT-29 cells were seeded in 24-well plates at a density of 8 × 10⁴ cells per well and incubated overnight at 37 °C in a 5% CO₂ atmosphere. Once the cells reached 80–90% confluency, the medium was replaced, and treatments were administered with free TQ, grind blank pectin beads, and grind TQ-PB. TQ was dissolved in DMSO and applied at concentrations ranging from 50 to 400 µg/mL, while TQ-PB were used at equivalent TQ concentrations. Blank pectin beads were applied at amounts corresponding to the number of beads used in the loaded formulations to assess their standalone effect. The half-maximal inhibitory concentration (IC50) was determined based on the resulting cell viability.

After the treatment period, the culture media were aspirated, and the cells were gently washed three times with 300 μL of PBS to remove any remaining treatment. Following this, 90 μL of fresh culture medium and 60 μL of MTT solution (5 mg/mL in PBS) were added to each well and incubated for an additional 4 h to promote the formation of formazan crystals [37]. After incubation, the culture medium was removed, and 1 mL of dimethyl sulfoxide (DMSO) was added to each well to dissolve the formazan crystals. The solution was then agitated for 5 min to ensure complete dissolution. The percentage of cell viability was calculated using Equation (6) after measuring the absorbance at 570 nm with a plate reader (Multiskan Go, Thermo Fisher Scientific, Waltham, MA, USA). All measurements were performed in triplicate.
(6)$$Cell\;viability=\frac{A_{control}-A_{sample}}{A_{control}}\times 100\%$$
where A_sample_ represents the absorbance of treated samples and A_control_ represents the absorbance of the control group (cells without treatment).

### 2.13. Statistical Analysis

All experiments were performed in triplicate as independent experiments, and data were expressed as mean values with standard deviation (±SD). Statistical analysis was conducted using IBM SPSS Statistics (version 25), with a *p*-value less than 0.05 considered a significant difference. Differences between variables were subjected to significance analysis (compare means) utilising one-way analysis of variance (ANOVA). Significantly different means (*p* < 0.05) were identified by post hoc Tukey analysis. Design-Expert software (version 13.0.5.0; Stat-Ease, Inc., Minneapolis, MN, USA) was used for statistical analysis and optimisation processes using central composite design (CCD) approach-based response surface methodology (RSM) analysis. Prism version 10.4.0 (Boston, MA, USA) was used to generate graphs and conduct statistical analyses. Non-significant values are indicated by ‘ns’ (*p* > 0.05), while asterisks denote significance levels: * *p* ≤ 0.05, ** *p* ≤ 0.01, *** *p* ≤ 0.001, and **** *p* ≤ 0.0001. The results were analysed using one-way and two-way analysis of variance (ANOVA).

## 3. Results and Discussion

To optimise the formulation of TQ-pectin beads for oral colonic delivery, a central composite design (CCD) based on response surface methodology (RSM) was employed to assess the interactions between four key preparation parameters and seven response variables. This study evaluated the influence of the pectin concentration (A), voltage (B), BSO ratio (C), CaCl_2_ concentration (D) on particle size (R1, mm), sphericity (R2, *w*/*w*), EE (R3, %), water uptake SGF (R4, %), water uptake SIF (R5, %), water uptake SCF (R6, %), and erosion (R7, %). The fit summary and the equation of each model are reported in Table 2.

### 3.1. The Effects of Formulation Variables on Particle Size, Sphericity, and EE

#### 3.1.1. Particle Size

BSO (Figure 2B,C) ratio showed a strong negative effect on particle size, with the smallest size achieved at a BSO ratio of approximately 1.5 mm. Beyond this ratio, the particle size increased, likely due to an increase in solution viscosity as oil content increased, which has been observed in previous studies [38]. High-viscosity solutions hinder the electrospray process, making it difficult to overcome surface tension, leading to larger droplets and a shift to a dripping mode [39]. Additionally, pectin concentration and voltage intensity also significantly affected the particle size, as illustrated in Figure 2C. As pectin concentration increased, the solution became more viscous, leading to larger particles. High voltage levels failed to focus the electric field at the needle tip, which explains the reduced effect of voltage on particle size [40]. A similar effect was seen with the interaction between pectin concentration and BSO ratio (Figure 2B), indicating that increased oil content, along with pectin, increases viscosity and limits the voltage’s impact on particle size formation.

#### 3.1.2. Sphericity

Pectin concentration exhibited a negative effect on the sphericity of the beads (Figure 2D,F). As the concentration of pectin decreased, the beads increasingly approached a spherical shape. This effect can be attributed to the limitations of the electrospray technique when processing viscous liquids. Increased pectin concentrations led to higher solution viscosity, which can hinder the voltage from effectively overcoming surface tension at the needle tip, resulting in beads with tails and reduced sphericity. In contrast, the BSO ratio had an insignificant impact on bead sphericity, as indicated by a *p*-value of 0.75 (Figure 2F).

Moreover, pectin, as a polymer, exhibits lower cross-linking potential compared to other polymers such as alginate or Arabic gum [41], which contributes to the reduced sphericity of beads when pectin concentrations approach 5%. Several studies have addressed these limitations by combining pectin with other polymers to enhance its physicochemical properties. For instance, one study utilised alginate as a co-polymer with pectin to produce spherical beads for colonic drug delivery using the electrospray method [42]. Another study optimised pectin bead formulations for agrochemical loading by adjusting pectin concentrations to achieve distinct bead morphologies. In a study by Pavithran et al. (2021), the morphology of pectin beads was found to vary based on pectin concentration. At 4%, the beads exhibited a flat and round shape, while at 6%, they became spherical with a smooth surface. Increasing the concentration to 8% resulted in pear-shaped beads, and at 10%, the beads developed a tear-shaped form with a smooth surface [43]. This consistency in bead morphology is crucial for achieving the desired mechanical strength and stability, ensuring that the beads can withstand varying conditions of the gastrointestinal tract while maintaining their integrity for targeted drug delivery [44]. Conversely, increasing the concentration of CaCl_2_ in conjunction with pectin led to a negative interaction effect on beads sphericity, as illustrated in Figure 2E. This effect may be attributed to the rapid cross-linking induced by the high CaCl_2_ concentration, which occurs immediately as the deformed jetted droplets of the highly viscous pectin solution enter the cross-linking bath.

#### 3.1.3. Encapsulation Efficiency (EE)

The EE% was negatively affected by the BSO ratio (Figure 2H), possibly due to oil leakage from the microbead matrix during the cross-linking process, as well as the limited loading capacity of the pectin matrix. This observation was supported by an increase in EE% with higher concentrations of pectin, as illustrated in Figure 2H. The voltage intensity demonstrated a positive effect on EE%, attributed to the increased surface area of the smaller pectin beads produced at higher voltages (Figure 2I). The larger surface area enhances interaction with the surrounding cross-linking solution, facilitating quicker gelation and minimising the loss of TQ during preparation.

### 3.2. Effects of Formulation Variables on Water Uptake in SGF, SIF, SCF, and Erosion

Figure 3A illustrates the interaction effect between pectin and CaCl₂ concentrations. As the CaCl_2_ concentration increased, the hydrogel bead matrix became stronger and more compact, leading to a reduction in the porosity of the three-dimensional structure. Consequently, this resulted in a significant decrease in the water uptake percentage over time [45]. The reduced water uptake is attributed to the interaction between the COO^−^ groups of pectin and the divalent Ca^2+^ cations, forming strong cross-linkages [46]. A similar trend was observed for water uptake percentages in both SIF and SCF conditions (Figure 3C,F,I). The BSO ratio impacts the swelling behaviour of beads in different simulated fluids (Figure 3B,E,G). In SGF, BSO demonstrates a positive effect by limiting pectin shrinkage, affecting porosity and the ability of the medium to infiltrate the beads, leading to drug release (Figure 3B,C). This is because BSO creates cavities within the matrix, reducing the compactness and promoting medium infiltration. However, in SIF and SCF, an increase in BSO led to a negative effect on water uptake (Figure 3F,I). This reduction in swelling is attributed to the lower amount of pectin in the formulation, which weakens the swelling capacity of the beads.

Pectin typically exhibits a shrinkage effect in acidic environments, such as SGF, and swells in neutral or alkaline conditions, such as SIF and SCF. However, at higher BSO ratios, the reduced pectin content limits the ability to swell effectively in these media, thereby impacting drug release and water uptake behaviours under these conditions [46]. As the voltage influences the size of pectin beads, increasing the voltage intensity results in the production of smaller beads. The voltage factor demonstrated a negative effect on water uptake in SGF (Figure 3C) but a positive effect in simulated intestinal fluid (SIF) (Figure 3F). The smaller pectin beads exhibited reduced water uptake in SGF (acidic conditions) due to the tighter cross-linking pectin matrix and increased shrinkage, thereby limiting water penetration. In SIF and SCF, smaller beads absorb more water than larger beads due to their greater surface area available for interaction with the natural and alkaline surrounding medium, facilitating quicker swelling. Conversely, larger beads exhibit slower water absorption because of longer diffusion paths and reduced surface area exposure [47,48].

Smaller beads produced under higher voltage exhibit greater erosion (Figure 3J–L) due to their increased surface area, which enhances interaction with the surrounding medium and accelerates degradation. In contrast, a denser cross-linked pectin matrix, formed by higher concentrations of CaCl_2_ (Figure 3J), mitigates erosion by reducing porosity and fluid diffusion, thereby rendering the matrix more resistant to degradation. The effects of pectin concentration and BSO ratio on erosion are minimal compared to the significant influences of voltage and CaCl_2_ concentration (Figure 3L). This is likely due to the dominance of cross-linking density and surface area as the primary factors affecting erosion.

### 3.3. Optimisation and Validation

The optimised TQ-pectin beads formulation was meticulously selected based on critical factors such as maximising the sphericity, the water uptake in SCF, and EE, while minimising the water uptake in SGF and SIF. The chosen formulation included a pectin concentration of 5%, a voltage intensity of 5.04, a BSO ratio of 1 to 10 of pectin solution, and a CaCl concentration of 5.36%. The predicted values for the dependent variables aligned with the desired ranges for this formulation (average particle size of 2.07 mm, sphericity of 0.98, EE of 90.1%, water uptake in SGF of 293.88%, water uptake in SIF of 470.60%, water uptake in SCF of 753.38%, and erosion of 47.87%). The actual results showed an average bead size of 2.05 ± 0.14 mm with sphericity of 0.96 ± 0.05, EE 90.32 ± 1.04%, water uptake in SGF of 287.55 ± 10.14%, water uptake in SIF of 462.15 ± 12.73%, water uptake in SCF of 772.41 ± 13.03%, and erosion of 45.23 ± 5.22%, which are closely matched to the predicted outcomes and within the range of point prediction with an error of less than 5%, validating the optimised TQ-PB formulation and the equations of each model. 

### 3.4. Surface Morphology Analysis of TQ-PB

Figure 4 presents the TQ-pectin beads, captured through optical digital imaging (Figure 4A,B) and SEM imaging (Figure 4C,D), highlighting their overall appearance and surface morphology. The beads exhibit a uniformly spherical shape, indicative of a well-controlled bead formation process using the electrospray technique. During preparation, the beads had a larger particle size of 2.88 ± 0.12 mm, which decreased to 2.05 ± 0.14 mm after drying due to moisture loss. The surface appears slightly rough, with noticeable undulations and small micropores. This texture suggests that, while predominantly smooth, the surface possesses micro-roughness or fine structuring, likely resulting from the interaction between pectin molecules and calcium ions during gelation. These micropores may affect the water uptake capacity and drug release kinetics of the beads. 

### 3.5. FTIR Analysis

The FTIR spectra of pectin, TQ, blank pectin beads, and TQ-PB show distinct shifts in characteristic peaks, indicating successful encapsulation and molecular interaction between TQ and pectin (Figure 5). For pectin, the broad peak observed at 3251.98 cm^−1^ corresponds to O-H stretching, while the bands at 1732.08 cm^−1^, 1631.78 cm^−1^, and 1371.39 cm^−1^ are attributed to C=O stretching, COO- asymmetric stretching, and C-H bending vibrations, respectively [49]. TQ shows prominent peaks at 2916.37 cm^−1^ and 2854.65 cm^−1^ (C-H stretching), 1627.92 cm^−1^ (C=O stretching), and 1365.6 cm^−1^ (aromatic C=C) [50]. In blank pectin beads, the broadening of the O-H peak at 3338.78 cm^−1^ and the shift of the C=O peak to 1635.42 cm^−1^ indicate hydrogen bonding between the pectin matrix and encapsulated components [51]. In the case of TQ-PB, the shifts observed in the hydroxyl region at 3361.93 cm^−1^, 2918.26 cm^−1^, and 2864.55 cm⁻^1^ via C-H stretching and at 1624.44 cm^−1^ in the C=O region suggest strong intermolecular interactions between pectin and TQ, confirming encapsulation.

The FTIR results indicate that pectin acts as an effective encapsulation material for TQ, forming strong hydrogen bonds, as seen from the shifts in peak positions in both PB and TQ-PB spectra. These results are consistent with previous studies that demonstrated the ability of pectin to form stable complexes with hydrophobic drugs through hydrogen bonding and electrostatic interactions, as highlighted by [49,51], who reported similar shifts in pectin-encapsulated curcumin. A study also observed comparable FTIR shifts during TQ encapsulation in chitosan nanoparticles, indicating that the hydrogen bonding observed in the current study effectively stabilises the drug in the polymer matrix [52]. 

### 3.6. Powder X-Ray Diffraction (PXRD) Analysis

The PXRD technique enables assessment of the prepared beads in terms of crystallinity and confirms whether the active ingredient is successfully entrapped within the beads. The PXRD results for electrosprayed beads are presented in Figure 6. Notably, the XRD patterns of both blank and TQ-PB prepared by electrospray technique display no sharp peaks, unlike the sharp peaks observed in crystalline thymoquinone. This absence of sharp peaks suggests that the active ingredient is homogeneously dispersed within the pectin polymer network, facilitating effective thymoquinone encapsulation and making the beads suitable for potential sustained release. Consistent with previous studies [53,54,55], this beading process promotes amorphisation of the active ingredient, enhancing entrapment and encapsulation.

### 3.7. Elastic-Plastic Behaviour and Compression Analysis

The recovery plot (Figure 7) demonstrates that TQ-PBs exhibit both elastic and plastic behaviour in their dry and wet forms, respectively, under compression. The percentage recovery of TQ-PB reflects their response to progressive deformation over time, such as 10% deformation at 0 s, 20% at 900 s, 30% at 1800 s, 40% at 2700 s, and 50% at 3600 s. This progression provides insights into the beads’ behaviour under varying levels of compression, as well as their adaptability to different gastrointestinal (GI) environments.

In dry form, TQ-PB initially showed high recovery, with 99.52% at 10% deformation. However, recovery decreased as deformation increased, dropping to 66.13% at 50% deformation. In contrast, wet beads displayed a lower initial recovery of 46.5% at 10% deformation, which declined more sharply to 34.08% at 50% deformation. This difference indicates that water uptake significantly increases plastic (permanent) deformation in the wet beads, reducing their capacity for full recovery under compression, as shown in Figure 8. Overall, TQ-PB beads primarily resist deformation elastically when dry but transition to plastic deformation upon water absorption in wet conditions, resulting in structural breakdown. This recovery trend is important for understanding the beads’ behaviour in the pharmaceutical industry and in the GI tract, where it impacts the release profile of encapsulated TQ.

### 3.8. Drug Release Studies and Release Kinetics of TQ

The release profile of TQ from the beads was evaluated across three simulated physiological media, namely, SGF (pH 1.2 ± 0.1), SIF (pH 6.8 ± 0.1), and SCF (pH 7.4 ± 0.1), as illustrated in Figure 9A. In SGF, TQ release was minimal, with only 8.13 ± 1.94% released after 2 h. This limited release suggests that the pectin encapsulation effectively shields the drug in acidic environments, likely due to the contraction of the pectin hydrogel at low pH [46].

Upon transitioning to SIF, a gradual increase in TQ release was observed, reaching 18.32 ± 3.21% after 2 h and 29.35 ± 3.65% after 4 h. This is attributed to the alkaline conditions in SIF, which promote swelling of the pectin matrix, enhancing drug diffusion. Subsequently, the release rate markedly accelerated in SCF, achieving 49.76 ± 2.54% after 1 h and 94.43 ± 2.4% after 3 h, with sustained release continuing over 24 h. This behaviour underscores the design of the pectin beads, which are tailored to degrade or swell selectively in the colonic environment in the presence of pectinase enzymes, facilitating targeted release of TQ in the colon [56].

To further elucidate the release kinetics of TQ from the pectin beads (TQ-PBs), various models, including zero-order, first-order, Higuchi, Korsmeyer–Peppas, and Hixson–Crowell models, were applied, as shown in Figure 9B. The correlation coefficient (R^2^) was used to assess the fit of each model. The zero-order model exhibited the highest determination coefficient (R^2^ = 0.9913), indicating that TQ release follows a zero-order kinetic pattern, characterised by a constant release rate over time, independent of drug concentration. This is further supported by the linearity of the zero-order release plot.

### 3.9. Stability Study

The stability study demonstrated that TQ-pectin beads underwent a reduction in particle size after 2 and 4 weeks of storage at 40 °C ± 2 °C and 25% RH ± 5% RH, as shown in Figure 10A. The particle size decreased by approximately 10.05% after 2 weeks and 17.50% after 1 month of storage. Additionally, the encapsulated TQ content diminished significantly over time, with a reduction of 5.41% at 2 weeks and 6.59% after 1 month, as reported in Figure 10B. The shrinkage of the beads during storage, as presented in Figure 10C,D, may be attributed to weight loss in the harsh environmental conditions of the stability test, likely due to the reduction in BSO content, which accounts for the observed decrease in both size and TQ content. The porous surface structure facilitates the diffusion of the encapsulated compound and increases its exposure to atmospheric oxygen, which depends on pore size [57]. Coating these micropores with a protective layer has been shown to enhance the stability of such formulations, as reported previously [58,59,60].

### 3.10. Cytotoxicity Study

The cytotoxic effects of free TQ, TQ-PB, and blank pectin beads on HT-29 cell lines were evaluated using the MTT assay over a 24 h period. Figure 11 demonstrates that HT-29 cell viability was influenced by the concentration of TQ, with blank pectin beads exhibiting significantly lower cytotoxicity compared to both free TQ and TQ-PB. Similarly, a dose-dependent cytotoxicity observed in this study is supported by a previous study, which noted a comparable response of TQ against HT-29 cells [61]. Over the concentration range of 50–400 μg/mL, the half-maximal inhibitory concentration (IC50) was determined to be 56.24 ± 2.8 μg/mL for free TQ, while for TQ-PB, it was significantly higher at 80.59 ± 2.2 μg/mL. The variation in IC50 values between free TQ and TQ-PB is likely due to the controlled release of TQ from the pectin matrix, which degrades in the presence of the pectinase enzyme. These findings are consistent with prior research by Sookkasem et al. (2017), who reported higher cytotoxicity for free curcumin compared to curcumin encapsulated in pectin beads in a colon-targeting delivery system [56]. The moderate cytotoxicity observed for the blank pectin beads is consistent with previous findings on pectin-based delivery systems. For instance, Chittasupho et al. (2013) demonstrated that pectin nanoparticles conjugated with methotrexate exhibited moderate cytotoxic effects against hepatic cancer cells, especially at elevated pectin concentrations [62]. This correlation highlights the influence of pectin content in driving cytotoxic activity, even in the absence of an active therapeutic agent.

## 4. Conclusions

This study successfully developed and optimised TQ-PBs for targeted colon-specific delivery using the electrospray technique. The optimised formulation demonstrated favourable particle size, sphericity, and EE, along with controlled water uptake and erosion in simulated gastrointestinal environments. The TQ-PB exhibited a controlled and targeted release profile, with minimal release in the gastric and intestinal phases and accelerated release in the colonic phase, making it suitable for colon-specific delivery. Moreover, stability studies also showed acceptable retention of TQ content over time, further supporting its viability for extended use. In addition, TQ-PB formulation represents a highly effective strategy for overcoming the bioavailability challenges of TQ and advancing its application through colorectal cancer targeting. Hence, TQ-PBs demonstrate strong potential as a promising oral drug delivery system for colorectal cancer therapy.

## Figures and Tables

**Figure 1 pharmaceutics-16-01460-f001:**
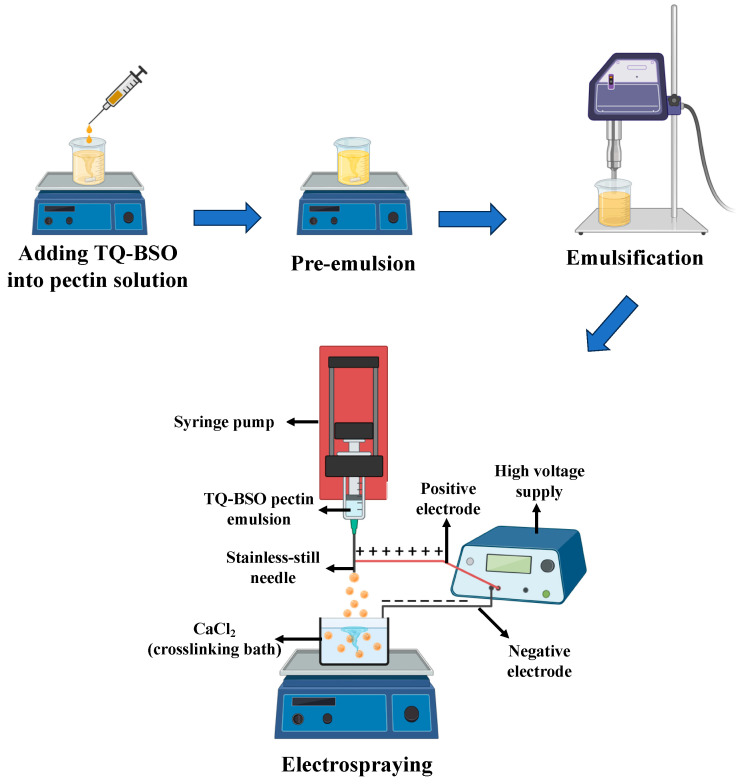
Schematic diagram of beads preparation by the electrospray method.

**Figure 2 pharmaceutics-16-01460-f002:**
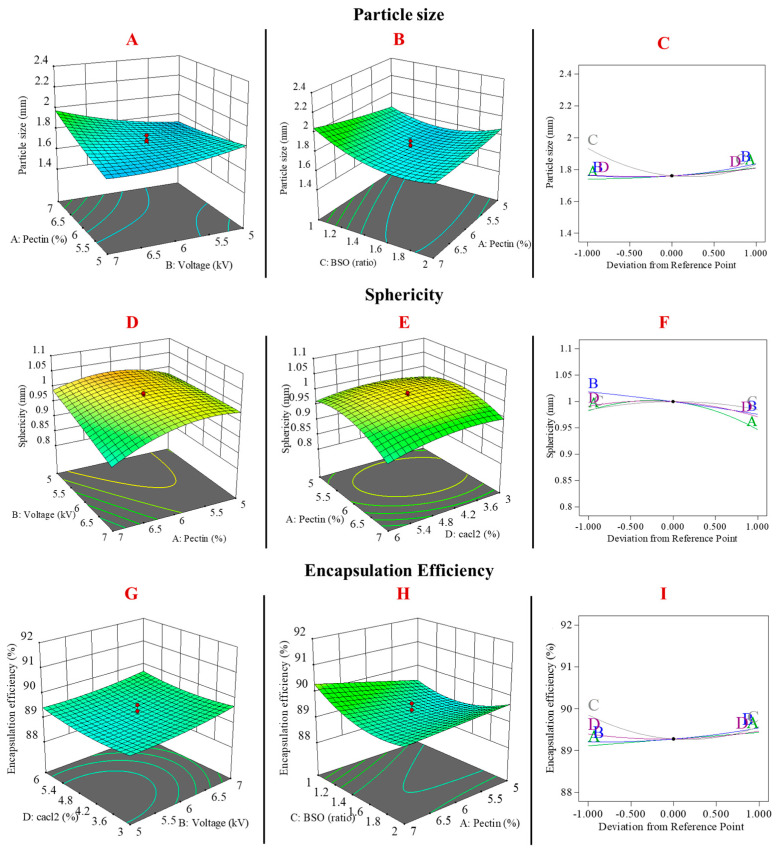
Illustration of the relationship between the design factors with particle size (nm) (**A**–**C**), sphericity (**D**–**F**), and encapsulation efficiency (%) (**G**–**I**) as the response variables.

**Figure 3 pharmaceutics-16-01460-f003:**
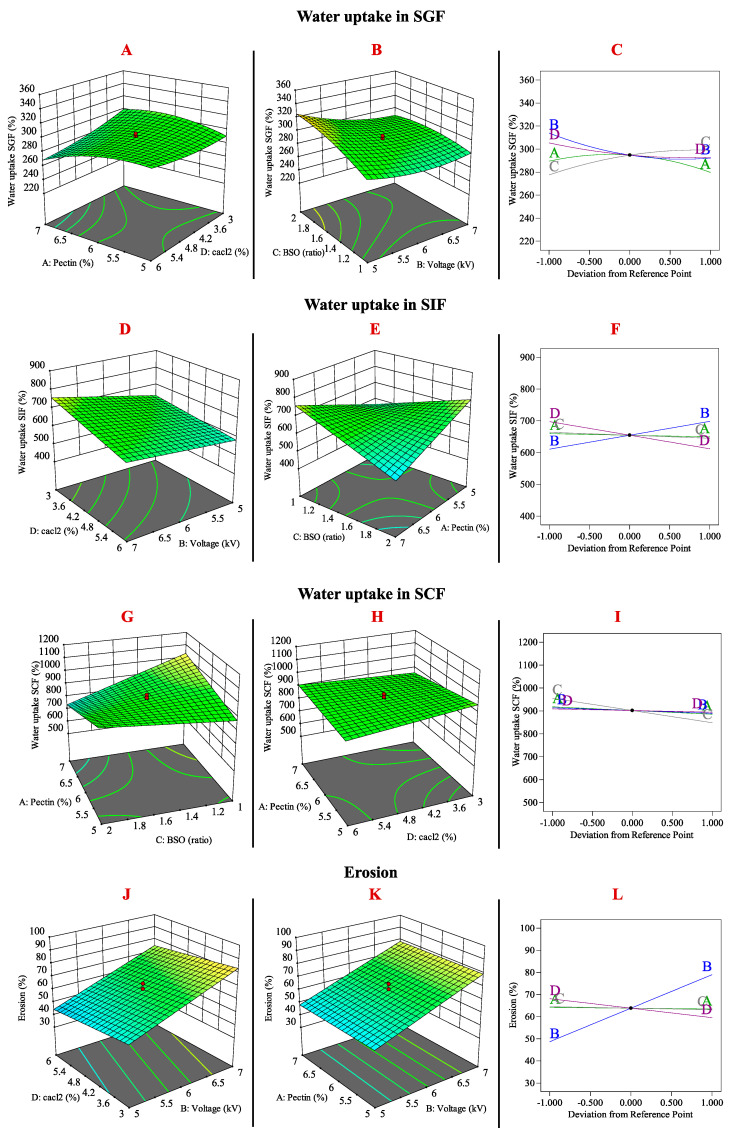
Illustration of the relationship between the design factors with water uptake in SGF (**A**–**C**), SIF (**D**–**F**), and SCF (**G**–**I**) and the erosion (**J**–**L**) of the TQ-PB as the response variables.

**Figure 4 pharmaceutics-16-01460-f004:**
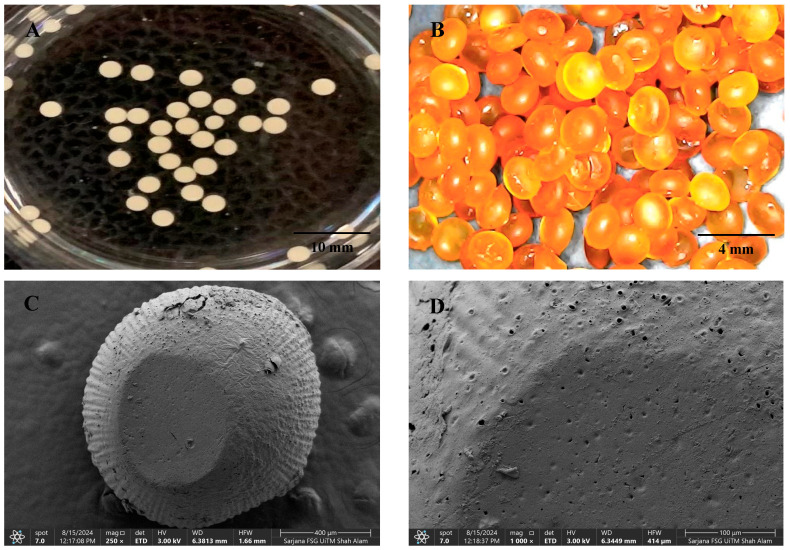
Optical images of TQ-pectin beads during the cross-linking process (**A**), optical images of dry TQ-PB (**B**), and SEM images of TQ-PB (**C**,**D**).

**Figure 5 pharmaceutics-16-01460-f005:**
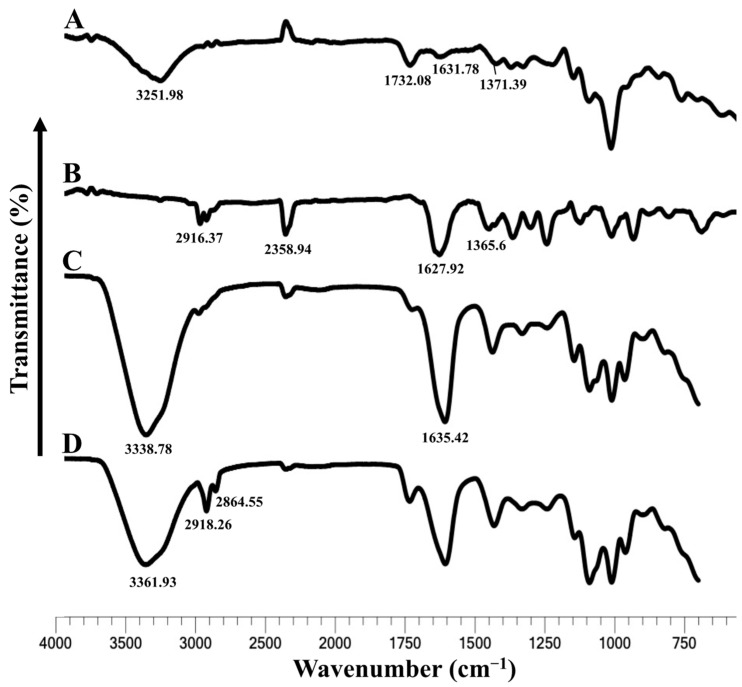
Presents the FTIR spectrum of pectin (A), thymoquinone (B), blank PB (C), and TQ-PB (D).

**Figure 6 pharmaceutics-16-01460-f006:**
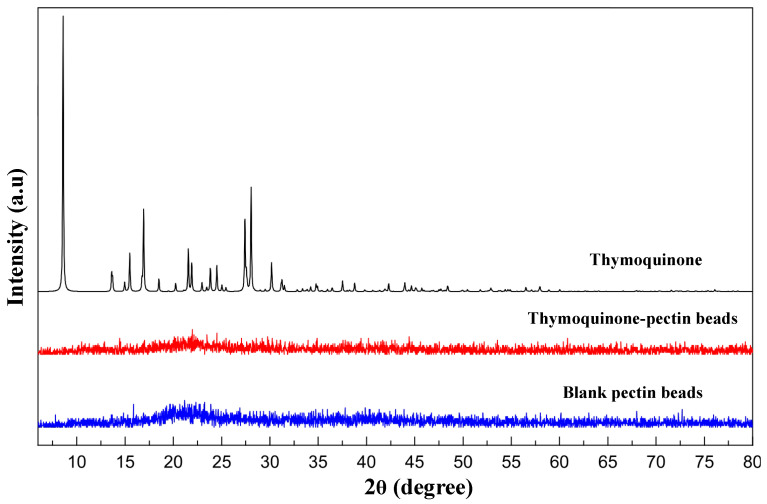
XRD patterns of TQ, blank PB, and TQ-PB.

**Figure 7 pharmaceutics-16-01460-f007:**
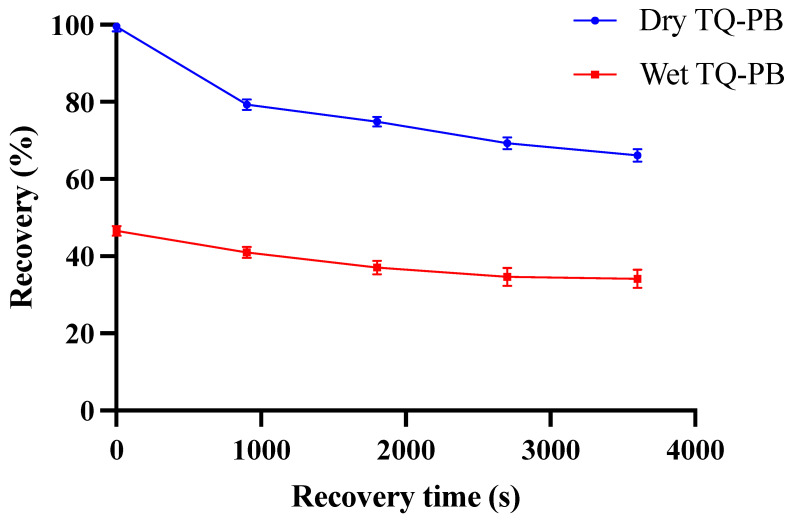
Compression (force (N) versus time in second) of dry and wet TQ-PB (upper-right) to various percentage deformations (10–50%) and their elastic–plastic behaviour (bottom-left).

**Figure 8 pharmaceutics-16-01460-f008:**
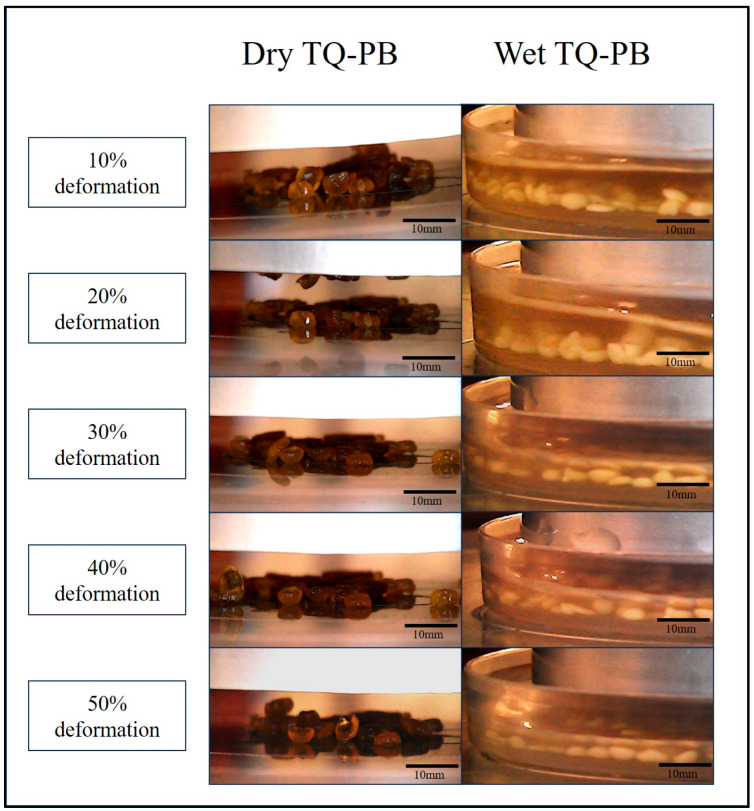
Optical images of dry and wet TQ-PB at various deformation levels (10–50%).

**Figure 9 pharmaceutics-16-01460-f009:**
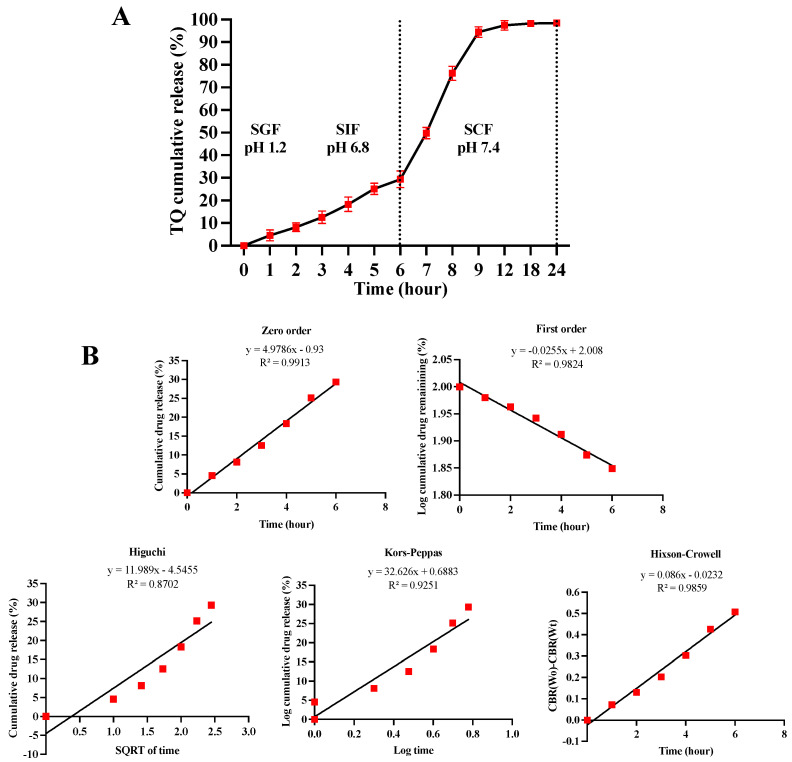
Illustration of the cumulative release profile of TQ from pectin beads in three different body mimicking fluids: SGF (pH 1.2 ± 0.1) for 2 h, SIF (pH 7.4 ± 0.1), and SCF (pH 6.8 ± 0.1) until complete TQ release (**A**), and the release kinetic models of the TQ (**B**) zero-order model, first-order model, Higuchi model, Korsmeyer–Peppas, and Hixson–Crowell model.

**Figure 10 pharmaceutics-16-01460-f010:**
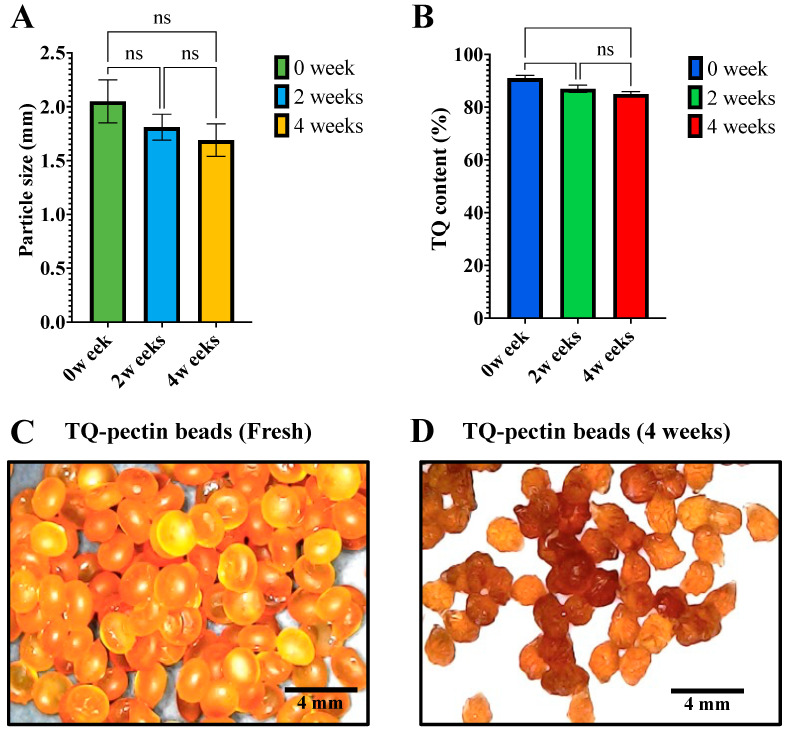
Illustration of the changes in particle size (**A**), TQ content (**B**), and the shape of TQ-pectin beads freshly prepared (**C**) and over the storage period at 40 °C ± 2 °C and 25% RH ± 5% RH (**D**).

**Figure 11 pharmaceutics-16-01460-f011:**
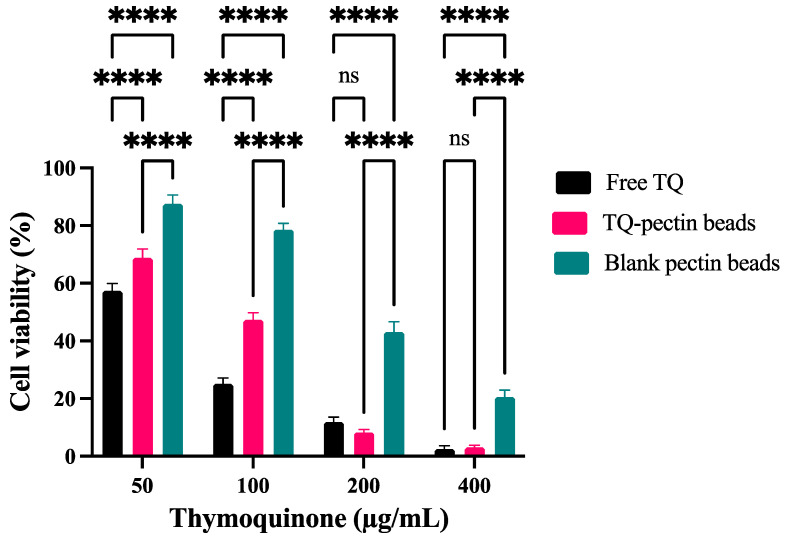
Cell viability of HT-29 cell lines after 24 h of incubation with free TQ, TQ-PB, and blank pectin beads.

**Table 1 pharmaceutics-16-01460-t001:** Independent and dependent variables of the experimental design.

Factor	Name	Unit	Low Level	High Level	−alpha	+alpha
A	Pectin concentration	%	5	7	4	8
B	Voltage	kV	5	7	4	8
C	BSO ratio	*w*/*w*	1	2	0.5	2.5
D	CaCl_2_ concentration	%	3	6	1.5	7.5

**Table 2 pharmaceutics-16-01460-t002:** The fit summary and the equation of each model generated by the Design-Expert software.

Code	Model	*p*-Value	Lack of Fit	R^2^	Predicted R^2^	Adjusted R^2^
Particle size (mm)						
R1	Quadratic	<0.0001	0.57	0.96	0.92	0.94
Particle size (mm) = 1.76 + 0.0355A + 0.0358B − 0.0510C + 0.0214D + 0.0916 AB − 0.0638AC + 0.0201AD + 0.1024BC − 0.0115BD − 0.0288CD + 0.0149A2 + 0.0400B^2^ + 0.1222C^2^ + 0.0275D^2^						
Sphericity						
R2	Quadratic	<0.0001	0.08	0.97	0.95	0.96
Sphericity = +0.9999 − 0.018A − 0.0212B + 0.0004C − 0.0099D − 0.0231AB + 0.0240AC − 0.0085AD − 0.0139BC − 0.0181BD − 0.0139CD − 0.0353A^2^ − 0.0027B^2^ − 0.0156C^2^						
Encapsulation efficiency (EE%)						
R3	Quadratic	<0.0001	0.57	0.84	0.69	0.78
EE (%) = 89.28 + 0.1612A + 0.1677B − 0.1417C + 0.0346D + 0.3250AB − 0.2894AC + 0.0688AD + 0.3644BC − 0.0275BD − 0.0994CD − 0.0033A^2^ + 0.0989B^2^ + 0.4367C^2^ + 0.1555D^2^						
Water uptake in SGF (%)						
R4	Quadratic	<0.0001	0.73	0.97	0.94	0.96
Water uptake in SGF (%) = 294.74 − 4.88A − 10.85B + 10.68C − 6.21D − 1.36AB + 6.53AC − 9.20AD − 7.49BC − 6.29BD + 2.00CD − 10.17A^2^ + 8.31B^2^ − 6.52C2 +4.36D^2^						
Water uptake in SIF (%)						
R5	2FI	<0.0001	0.06	0.99	0.98	0.99
Water uptake in SIF (%) = 654.54 − 5.18A + 43.73B − 9.40C − 42.72D − 21.41AB − 100.74AC − 2.08AD − 22.90BC − 16.72BD − 1.74CD						
Water uptake in SCF (%)						
R6	2FI	<0.0001	0.16	0.82	0.70	0.78
Water uptake in SCF (%) = 901.63 − 16.18A − 11.94B − 53.51C − 7.23D − 41.86AB − 95.67AC + 31.82AD − 70.46BC + 13.80BD − 7.58CD						
Erosion (%)						
R7	Linear	<0.0001	0.22	0.97	0.97	0.97
Erosion (%) = 63.83 − 0.3789A + 15.15B − 0.6988C − 4.29D						

## Data Availability

The original contributions presented in the study are included in the article; further inquiries can be directed to the corresponding authors.

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
