# Peer review of "Thymoquinone Pectin Beads Produced via Electrospray: Enhancing Oral Targeted Delivery for Colorectal Cancer Therapy"

_pharmaceutics, 2024, doi:10.3390/pharmaceutics16111460_

Round 1

Reviewer 1 Report

Comments and Suggestions for Authors

The Authors describe the preparation of pectine beads loading an anticancer drug for a targeted colonic delivery. The literature already describe the potential application of this polymer for colon delivery. Moreover, several studies highlight the potential benefit of pectine based multi-stimuli carriers for intestinal (colon) delivery by using the properties of the polymer, able to release a drug as function of pH, time and enzymes. The Authors should explain the novelty/originality of their study.

Does the type of pectine (molecular weigh, composition must be defined) and the technique use impact on the production of the carriers and drug/carrier stability, drug release and loading? What is the effect of the type of oil use on these parameters as well? Please, explain/justify the choice of that polymer and that oil used. 

The Authors should define the carriers. Are they matrix or capsule like systems? A SEM of cross-section could demonstrate the type of system produced.

Author Response

Reviewer 1:

Comment (1): The Authors describe the preparation of pectine beads loading an anticancer drug for a targeted colonic delivery. The literature already describe the potential application of this polymer for colon delivery. Moreover, several studies highlight the potential benefit of pectine based multi-stimuli carriers for intestinal (colon) delivery by using the properties of the polymer, able to release a drug as function of pH, time and enzymes. The Authors should explain the novelty/originality of their study.

Authors’ response:

We sincerely appreciate your valuable comments and insights, the authors hypothesized that the novelty of this study is utilizing the electrospray technique to fabricate pectin beads with precisely controlled sizes could enhance their behaviours in the gastrointestinal tract. By adjusting the bead diameter, we aimed to modulate water absorption and erosion rates, thereby achieving a controlled release profile for encapsulated TQ that enhances targeting specifically to the colon. The authors was mentioned on that in lines 109-119.

Comment (2): Does the type of pectine (molecular weigh, composition must be defined) and the technique use impact on the production of the carriers and drug/carrier stability, drug release and loading? What is the effect of the type of oil use on these parameters as well? Please, explain/justify the choice of that polymer and that oil used.

Authors’ response:

The pectin citrus used in this study is a pectin with a molecular weight (MW) >100k g/mol (Product No. 26234-05), as mentioned in lines 128 and 129.

The authors noted that the electrospray technique allows precise control over bead diameter, which enables optimization of water absorption and erosion rates. This, in turn, facilitates a controlled release profile for encapsulated TQ, enhancing targeted delivery specifically to the colon. The authors was mentioned on that in lines 109-113.

The authors choose Black seed oil due to its a natural source of thymoquinone with a thymoquinone solubility exceeding 500 mg/mL. So it was employed as a solvent of thymoquinone to replace toxic organic solvents and increase the drug loading capacity of the developed thymoquinone pectin beads, as mentioned in lines 113-116.

Comment (3): The Authors should define the carriers. Are they matrix or capsule like systems? A SEM of cross-section could demonstrate the type of system produced.

Authors’ response:

The authors already highlighted the type of the carriers as mentioned on lines 117-120.

Reviewer 2 Report

Comments and Suggestions for Authors

In this manuscript, the authors reported electrospray-fabricated thymoquinone pectin beads, and demonstrated their performance for colorectal cancer therapy. This work seems to be useful for this field. However, the following problems should be addressed before further consideration of publication:

1. The abstract should be revised with simplified description and formal format. The innovation can be demonstrated especially the superiority when compared with other researches.

2. All the figures need to be revised with consistent layout and style to improve the readability.

3. In the Introduction section, typical examples of drug delivery and targeted therapy should be introduced. The recent advances should be contained including: 10.1021/acsami.1c16859, 10.1002/EXP.20230037.

4. Material characterization of the samples are quite simple, and detailed analysis of their physical properties are suggested to be added such as XRD, XPS, density, porosity, elasticity modulus.

5. The authors have provided results of drug release as shown in Figure 6a. However, the complete drug release process in these fluids can be separately added for better comparison. Besides, the diversity and mechanisms should also be stated.

6. I’m confused with the experimental details of cell viability analysis. Considering the particle size, the co-incubation process should be conducted using grinded microscale beads.

7. In the manuscript, the depth could be improved if the authors provided some insights of micro-/nano interactions of the drug release and degradation process.

8. The references format should be checked due to some errors.

Author Response

Reviewer 2:

In this manuscript, the authors reported electrospray-fabricated thymoquinone pectin beads, and demonstrated their performance for colorectal cancer therapy. This work seems to be useful for this field. However, the following problems should be addressed before further consideration of publication:

Comment (1): The abstract should be revised with simplified description and formal format. The innovation can be demonstrated especially the superiority when compared with other researches.

Authors’ response:

Thank you for your valuable comments and insights. The authors have revised the abstract accordingly.

Comment (2): All the figures need to be revised with consistent layout and style to improve the readability.

Authors’ response:

The authors have modified the figures of the study accordingly.

Comment (3): In the Introduction section, typical examples of drug delivery and targeted therapy should be introduced. The recent advances should be contained including: 10.1021/acsami.1c16859, 10.1002/EXP.20230037.

Authors’ response:

The authors have improved the introduction of the study based on your comments, lines 68-74.

Comment (4): Material characterization of the samples are quite simple, and detailed analysis of their physical properties are suggested to be added such as XRD, XPS, density, porosity, elasticity modulus.

Authors’ response:

The authors already added the XRD analysis for the thymoquinone, blank pectin beads, and thymoquinone pectin beads, in the lines 230-234 and 466-478.

As well as the Authors alredy tested the texture and elasticity modulus of the thymoquinone pectin beads, as mentioned in line 235-246 and line 479-502.

Comment (5): The authors have provided results of drug release as shown in Figure 6a. However, the complete drug release process in these fluids can be separately added for better comparison. Besides, the diversity and mechanisms should also be stated.

Authors’ response:

The drug release profile of the optimized thymoquinone pectin beads was evaluated sequentially by first immersing the beads in simulated gastric fluid for two hours, followed by transferring them to simulated intestinal fluid for four hours to complete the release profile. Additionally, release kinetics were analyzed to model the full drug release profile of the thymoquinone pectin beads, applying various kinetic models including zero-order, first-order, the Higuchi equation, Korsmeyer-Peppas, and Hixson-Crowell models.

Comment (6): I’m confused with the experimental details of cell viability analysis. Considering the particle size, the co-incubation process should be conducted using grinded microscale beads.

Authors’ response:

The authors have highlighted the method applied for the MTT assay, ensuring the formulations were thoroughly ground before conducting the cytotoxicity study.

Comment (7): In the manuscript, the depth could be improved if the authors provided some insights of micro-/nano interactions of the drug release and degradation process.

Authors’ response:

The authors have modified the manuscript followed by your comment in line 68-74.

Comment (8): The references format should be checked due to some errors.

Authors’ response:

The authors have reviewed the reference style for accuracy and consistency.

Round 2

Reviewer 1 Report

Comments and Suggestions for Authors

There are no further comments for the Authors.

Author Response

Thank you for your kind report.

Reviewer 2 Report

Comments and Suggestions for Authors

It is my pleasure to review this manuscript, and the revisions have been thoroughly checked. However, minor revisions are needed to address some issues according to the comments in last revision.

1. In the Introduction, typical example of drug delivery and targeted therapy should be introduced. This reference was not included, which was also useful in this case: 10.1021/acsami.1c16859.

2. The author has added XRD in Figure 6, and the standard diffraction peaks can be noted. Besides, BET analysis also can be added to show its porosity and physical properties.

Author Response

It is my pleasure to review this manuscript, and the revisions have been thoroughly checked. However, minor revisions are needed to address some issues according to the comments in last revision.

Comment (1). In the Introduction, typical example of drug delivery and targeted therapy should be introduced. This reference was not included, which was also useful in this case: 10.1021/acsami.1c16859.

Authors response: Thank you for your insightful feedback. The introduction has been revised accordingly to include a discussion of typical examples of drug delivery and targeted therapy, line 66-70

Comment (2). The author has added XRD in Figure 6, and the standard diffraction peaks can be noted. Besides, BET analysis also can be added to show its porosity and physical properties.

Authors response: Thank you very much for your valuable suggestions. We appreciate your recommendation to include BET analysis to demonstrate the porosity and physical properties. However, BET analysis and mechanical characterization fall outside the primary scope of this study, as our main objective is to produce pectin beads of controlled size for targeted delivery of thymoquinone to the colon. The release profile and kinetics modeling confirm the effectiveness of the bead formulation in achieving this targeted release. Additionally, our lab does not currently have the facilities for comprehensive physical and mechanical characterization. Future work will focus on the detailed mechanical properties of these beads as we further refine this delivery system.